# Standardized Thalassotherapy Versus Conventional Rehabilitation in Post-Traumatic Patients: Clinical, Biochemical, and Quality-of-Life Outcomes

**DOI:** 10.3390/healthcare14010024

**Published:** 2025-12-21

**Authors:** Mihaela Mihai, Nica Sarah Adriana, Brindusa Ilinca Mitoiu, Liliana Sachelarie, Roxana Nartea

**Affiliations:** 1Corpore Sano Sanatorium Techirghiol, Faculty of Medicine, University of Medicine and Pharmacy “Carol Davila”, 050474 Bucharest, Romania; 2Faculty of Medicine, University of Medicine and Pharmacy “Carol Davila”, 050474 Bucharest, Romania; sarah.nica@umfcd.ro (N.S.A.); roxana.nartea@umfcd.ro (R.N.); 3Preclinical Sciences Department, Faculty of Medicine, Apollonia University, 700511 Iasi, Romania

**Keywords:** inflammation, marine mud, post-traumatic recovery, quality of life, rehabilitation, thalassotherapy

## Abstract

**Background:** Thalassotherapy, which combines seawater, marine mud, and maritime climate, has been traditionally used to support musculoskeletal recovery. Its thermal, biochemical, and mechanical properties may enhance tissue healing and modulate inflammation. This study aimed to evaluate the short-term clinical effects of a standardized two-week thalassotherapy program compared with conventional rehabilitation in post-traumatic patients. **Methods:** A matched controlled cohort study was conducted at the Corpore Sano Sanatorium, Techirghiol, Romania. Post-traumatic patients followed identical physiotherapy and hydrokinetic exercise routines; additionally, the thalassotherapy group received daily seawater baths, sapropelic mud applications, and exposure to marine aerosols. Pain levels, joint mobility, inflammatory status, and quality of life were assessed before and after the intervention. Adverse events and treatment tolerance were monitored throughout the study. **Results:** Patients undergoing thalassotherapy experienced more pronounced improvements in musculoskeletal function, pain relief, inflammatory balance, and quality-of-life outcomes compared with those receiving standard rehabilitation alone. Both interventions contributed to clinical progress, but the magnitude of change was consistently greater among patients treated with marine-based therapies. No adverse events or intolerance reactions were recorded, and all participants completed the program. **Conclusions:** Thalassotherapy may provide complementary short-term benefits in post-traumatic rehabilitation, enhancing functional recovery, symptom relief, and perceived well-being. However, due to the non-randomized design and short follow-up period, these findings should be interpreted cautiously. Further randomized studies with long-term outcomes are required to confirm the therapeutic role of thalassotherapy in modern rehabilitation practice.

## 1. Introduction

Thalassotherapy, defined as the therapeutic use of seawater, marine mud, algae, and the maritime climate, has a long tradition in medicine, being practiced for centuries in coastal regions to restore health and well-being [1]. In recent decades, its relevance has been reinforced by an increasing number of clinical and experimental studies that emphasize its multifactorial benefits, ranging from musculoskeletal recovery to improvements in metabolic and cardiovascular balance, as well as psychological well-being [2]. This broad applicability has positioned thalassotherapy as a complementary approach within modern rehabilitation medicine, bridging traditional natural therapies with contemporary evidence-based practices.

The therapeutic effects of thalassotherapy are based on complex interactions among the physical, chemical, and biological components of the marine environment [3]. Seawater is naturally enriched with essential minerals and trace elements, such as magnesium, calcium, potassium, and iodine, which contribute to remineralization, anti-inflammatory responses, neuromuscular relaxation, and regulation of metabolic and endocrine functions [4,5,6]. These elements can enter the body through cutaneous absorption during immersion or via inhalation of aerosols, which are particularly rich in ionic particles [7]. Such mechanisms promote systemic responses, including stimulation of thyroid activity through iodine uptake, regulation of cardiovascular function by potassium, and enhanced muscle relaxation through the actions of calcium and magnesium, thereby supporting both localized recovery and global physiological balance [8,9].

Marine muds and algae further extend these effects by providing bioactive molecules with therapeutic potential, including polyphenols, fucoidans, and carotenoids. These compounds are widely documented for their antioxidant, anti-inflammatory, and regenerative actions, which are capable of modulating cytokine activity, strengthening endogenous antioxidant defenses, and stimulating fibroblast proliferation. This collective effect favors tissue repair and functional recovery [10,11,12]. In parallel, the mechanical and thermal effects of immersion in warm seawater enhance circulation, facilitate venous and lymphatic return, and reduce joint and muscle stiffness. The maritime climate itself, enriched with negative ions and marine aerosols, contributes to improved respiratory function, autonomic regulation, and psychological relaxation, emphasizing the holistic character of thalassotherapy [5,13].

Taken together, these synergistic mechanisms explain why thalassotherapy can be successfully applied in diverse clinical contexts, particularly in rehabilitation programs. For musculoskeletal disorders and post-traumatic conditions, it has been shown to alleviate chronic pain, accelerate tissue healing, improve joint mobility, and reduce edema through hydrostatic and thermal mechanisms, while simultaneously supporting recovery at the biochemical level via marine-derived compounds [10,14,15]. Evidence from clinical studies supports these outcomes; for example, a quasi-experimental trial demonstrated that a two-week thalassotherapy program conducted in a seaside environment significantly improved balance, mobility, pain, and perceived well-being in post-stroke patients, thus underscoring the systemic dimension of its therapeutic potential [5].

By modulating adipokine secretion, downregulating pro-inflammatory cytokines like IL-6 and TNF-α, and reducing serum CRP, thalassotherapy provides systemic benefits that extend beyond metabolic regulation, encompassing cardiovascular protection and neuroendocrine balance [12,13,14,15,16]. Moreover, improvements in lipid and glucose metabolism, as well as favorable changes in body composition and oxidative stress markers, have been reported in patients undergoing structured programs of marine-based therapy, highlighting its preventive and rehabilitative roles. From this perspective, thalassotherapy emerges as an integrative therapeutic option that addresses multiple dimensions of health simultaneously, including musculoskeletal function, systemic metabolism, inflammatory regulation, and psychological well-being. These multifaceted benefits justify its inclusion in modern rehabilitation protocols, either as a complement to physiotherapy and pharmacological treatments or as a preventive measure in patients at risk of chronic disorders. In the context of post-traumatic recovery, however, its comparative effectiveness against conventional rehabilitation programs remains insufficiently documented.

Therefore, the present matched controlled cohort study, conducted at the Corpore Sano Sanatorium in Techirghiol, Romania, aimed to evaluate whether a standardized thalassotherapy protocol provides superior functional and biochemical outcomes compared with standard rehabilitation in post-traumatic patients.

Despite these promising findings, essential gaps persist in the current scientific literature. Most available studies investigate chronic musculoskeletal or mixed patient populations, while controlled comparisons between standardized thalassotherapy protocols and conventional rehabilitation specifically in post-traumatic patients remain scarce [9,14,17,18]. Moreover, only a limited number of studies have evaluated objective biochemical or inflammatory markers, such as CRP or cytokine profiles, to quantify systemic responses to marine-based therapies, despite indications that thalassotherapy may modulate inflammatory pathways [12,17,19]. Another major limitation is the lack of standardized intervention protocols regarding mineral composition, temperature, exposure duration, and procedural reproducibility, resulting in substantial heterogeneity across studies and reduced comparability [8,15,16].

Furthermore, high-quality European and Romanian clinical data remain scarce, despite the existence of longstanding rehabilitation centers such as Techirghiol, which provide unique mineral and climatic conditions with well-documented therapeutic potential [20].

These limitations reinforce the need for rigorously designed comparative studies that integrate clinical, biochemical, and quality-of-life outcomes within a standardized and reproducible thalassotherapy framework.

This study aimed to compare the short-term clinical, biochemical, and quality-of-life effects of standardized thalassotherapy versus conventional rehabilitation in post-traumatic patients.

## 2. Materials and Methods

### 2.1. Study Design

Over 2 years, a prospective matched-cohort study was conducted at the Corpore Sano Sanatorium in Techirghiol, Romania, to evaluate the short-term efficacy of thalassotherapy compared with standard rehabilitation in post-traumatic patients. The study received Institutional Ethics Committee approval (Approval No. 1947, 28 October 2023) and adhered to the ethical principles of the 2013 Declaration of Helsinki and the STROBE guidelines for observational research.

A total of 120 consecutive eligible patients with post-traumatic musculoskeletal conditions were enrolled. Patients were matched according to age, sex, type of injury, and baseline functional status, and subsequently allocated into two groups: Group A (n = 60), who underwent a standardized thalassotherapy-based rehabilitation program, and Group B (n = 60), who received conventional rehabilitation without marine components. Eligible participants were adults aged 25–75 years who had completed the acute phase of injury or orthopedic surgery and were referred for functional rehabilitation. Because the study was conducted in a single center, the sample size was determined by feasibility and no a priori power calculation was performed (Figure 1).

### 2.2. Participants

Eligible participants were adults aged 18–70 years who had completed the acute recovery phase following fractures, ligament injuries, or orthopedic surgeries and were medically stable to undergo functional rehabilitation. All participants provided written informed consent before inclusion.

Exclusion criteria included acute infections, decompensated heart failure, active malignancies, severe vascular disease, or any contraindications to hydrotherapy or marine exposure. Patients who did not complete the full two-week intervention or who had incomplete clinical data were excluded.

Participants were matched based on age, sex, type of injury, and baseline functional status, and allocated into two groups: Group A (n = 60): standardized thalassotherapy-based rehabilitation; Group B (n = 60): conventional rehabilitation without marine components.

The occurrence of adverse events, intolerance reactions, or early withdrawals was monitored throughout the study period, and none were reported.

### 2.3. Intervention Protocol

All participants followed a two-week rehabilitation schedule consisting of five sessions per week, conducted under medical supervision.

Patients in the thalassotherapy group (Group A) followed a multimodal marine-based rehabilitation program that included daily heated seawater baths lasting approximately 20 min at 36–38 °C, sapropelic mud applications applied for 15–20 min at 38–45 °C using peloids obtained from Lake Techirghiol, scheduled exposure to marine aerosols in controlled environments, and light to moderate hydrokinetic exercises performed in thermal seawater pools. This integrated protocol combined thermal, mechanical, and biochemical interventions designed to reduce inflammation, enhance peripheral circulation, promote muscle relaxation, and restore joint mobility.

In contrast, patients allocated to the control group (Group B) participated in a conventional rehabilitation program that comprised individualized kinesiotherapy, electrotherapy procedures, and therapeutic massage, without any marine-based components.

### 2.4. Outcome Measures

Clinical and biochemical parameters were assessed at two time points: baseline (Day 1) and immediately after completion of the program (Day 14). The evaluation protocol was multidimensional and included both primary and secondary outcomes. Pain intensity was measured using the Visual Analog Scale (VAS), while joint mobility was quantified through standardized goniometric techniques. Secondary outcomes focused on inflammatory status, assessed by serum C-reactive protein (CRP) concentrations, and on quality of life, evaluated using the validated Short Form-36 (SF-36) Health Survey. This instrument covers eight domains encompassing physical, emotional, and social dimensions of health, with total scores standardized on a 0–100 scale, where higher values reflect better perceived health and functional status. This comprehensive evaluation enabled an integrated appraisal of musculoskeletal recovery, inflammatory modulation, and psychosocial well-being following each intervention.

### 2.5. Statistical Analysis

Data were analyzed using SPSS version 27.0 (IBM Corp., Armonk, NY, USA). Descriptive statistics (means, standard deviations, and frequencies) were used to characterize the study population and summarize baseline clinical parameters. Comparative analyses were conducted using Student’s t-test for paired and independent samples to evaluate differences within and between groups. At the same time, one-way ANOVA was applied to assess mean changes across subgroups where applicable.

To identify predictors of rehabilitation outcomes, a multiple linear regression analysis was performed to examine the relationships between independent variables (treatment type, age, baseline functional status, and duration since injury) and dependent variables (pain reduction, improvement in joint mobility, CRP variation, and SF-36 score change). Regression analyses were considered the primary inferential method for evaluating the independent effect of treatment while adjusting for key covariates. The intragroup (baseline vs. post-intervention) comparisons are presented only as supplementary descriptive information. A repeated-measures ANOVA was applied to assess the interaction between treatment group and time, followed by Tukey post hoc tests for pairwise comparisons. The level of statistical significance was set at *p* < 0.05 for all analyses.

## 3. Results

### 3.1. Baseline Profile of the Study Population

The baseline characteristics of the study population are summarized in Table 1. The median age of participants in Group A was 46.8 years (IQR: 41–53), whereas Group B had a median age of 52.4 years (IQR: 48–57). This difference was statistically significant (*p* < 0.05), indicating that patients in Group B were, on average, older than those in Group A.

Regarding the environment of origin, Group A included 36 urban residents (60.0%) and 24 rural residents (40.0%), while Group B comprised 38 urban participants (63.3%) and 22 rural participants (36.7%). No significant difference was observed between the two groups (*p* = 0.68).

Socioeconomic status distribution was similar across groups, with 65% in the medium and 35% in the high socioeconomic status category in Group A, compared with 60% in the medium and 40% in the high in Group B (*p* = 0.52). Educational level also showed no significant differences between the groups: in Group A, 45% had completed high school, and 55% held a university degree, while in Group B the proportions were 48% and 52%, respectively (*p* = 0.74).

These results indicate that, aside from age, the two groups were comparable in terms of socioeconomic background, educational level, and living environment, supporting the validity of subsequent intergroup comparisons.

### 3.2. Intragroup Comparison (Baseline vs. Post-Intervention)

An intragroup analysis was performed to evaluate the evolution of clinical and quality-of-life parameters before and after the two-week rehabilitation period in both groups (Table 2).

Within-group comparisons revealed significant improvements across most clinical parameters following rehabilitation (Table 2). In Group A (thalassotherapy), joint mobility increased from 58.2 ± 12.5 to 75.6 ± 11.4 (*p* < 0.01), accompanied by a substantial reduction in pain intensity (VAS: 6.8 ± 1.5 to 3.9 ± 1.4, *p* < 0.01). Inflammatory activity also decreased significantly, with CRP levels declining from 4.6 ± 2.1 to 3.5 ± 1.8 mg/L (*p* < 0.01). Quality-of-life scores improved markedly, rising from 56.0 ± 14.0 to 67.8 ± 12.5 (*p* < 0.01).

In Group B (standard rehabilitation), improvements were also observed, albeit to a lesser extent. Joint mobility increased from 59.0 ± 13.1 to 66.8 ± 12.2 (*p* = 0.01), while pain scores decreased from 6.5 ± 1.6 to 5.2 ± 1.5 (*p* = 0.02). Quality of life improved significantly (from 54.5 ± 13.2 to 61.2 ± 12.8, *p* = 0.01). However, the reduction in CRP was not statistically significant (4.5 ± 2.0 to 4.1 ± 1.9 mg/L; *p* = 0.14).

Overall, both rehabilitation programs resulted in measurable clinical benefits, although thalassotherapy was associated with a greater magnitude of improvement across functional, inflammatory, and quality-of-life outcomes.

### 3.3. Intergroup Comparison (Group A vs. Group B)

An intergroup comparison was conducted to evaluate differences in clinical, inflammatory, and quality-of-life outcomes between post-traumatic patients undergoing thalassotherapy (Group A) and those receiving standard rehabilitation (Group B) after the two-week intervention (Table 3).

Table 3 shows that patients in the thalassotherapy group achieved significantly greater post-intervention improvements compared with those undergoing standard rehabilitation. Joint mobility was markedly higher in Group A (*p* = 0.01), and pain levels were significantly lower, with Group A demonstrating a more substantial reduction (*p* < 0.01). Inflammatory activity, as reflected by CRP values, was also considerably reduced in the thalassotherapy group (*p* = 0.02).

Although both groups showed improvements in quality-of-life scores, the thalassotherapy group obtained slightly better post-intervention values, though the difference did not reach statistical significance (*p* = 0.12).

Overall, these intergroup results indicate that thalassotherapy provides superior clinical benefits in post-traumatic rehabilitation, offering enhanced mobility recovery, more pronounced pain reduction, and meaningful modulation of inflammatory markers, outperforming standard rehabilitation programs.

### 3.4. Multifactorial Analysis (Repeated-Measures ANOVA)

To evaluate the combined influence of treatment type and time on clinical and biochemical outcomes, a repeated-measures ANOVA was performed with Group (thalassotherapy vs. standard rehabilitation) as the between-subject factor and Time (baseline vs. post-intervention) as the within-subject factor.

The analysis revealed significant main effects of Time for all evaluated parameters, indicating overall improvement from baseline across the entire cohort. A significant main effect of Group was also observed for each parameter, demonstrating the consistent superiority of the thalassotherapy program over standard rehabilitation.

A significant Group × Time interaction was observed across all parameters, indicating that the trajectory of improvement differed between the two groups, with the thalassotherapy cohort exhibiting more pronounced and accelerated improvement than the standard rehabilitation cohort. Post hoc Tukey comparisons supported these findings, showing significant within-group improvement in both cohorts (*p* < 0.01), with consistently higher effect sizes in the thalassotherapy group. Table 4 provides the detailed ANOVA outputs, including F-statistics, *p*-values, and partial η^2^ values for effect size.

Table 4 Results of the repeated-measures ANOVA assessing the effects of treatment (Group), time (baseline vs. post-intervention), and their interaction on clinical, biochemical, and quality-of-life parameters. All effects were statistically significant (*p* < 0.05), indicating superior improvement in the thalassotherapy group. Partial η^2^ values reflect effect sizes.

### 3.5. Exploratory Analyses: Correlations and Responder Profiles

To further explore the relationships between clinical and inflammatory parameters, exploratory analyses were performed, including correlation testing and responder profiles, better to characterize the impact of thalassotherapy, Table 5.

Table 5 illustrates the exploratory relationships between clinical improvement, inflammatory activity, and subjective well-being following thalassotherapy. A moderate positive correlation was found between pain reduction (Δ VAS) and CRP decrease (r = 0.44, *p* = 0.03), suggesting that the anti-inflammatory effect of marine-based therapy may contribute to its analgesic benefits. A stronger correlation was observed between mobility improvement and enhanced quality-of-life scores (r = 0.56, *p* < 0.01), emphasizing the close link between restored joint function and psychosocial recovery. A weaker but still significant association between CRP reduction and quality-of-life improvement (r = 0.39, *p* = 0.04) supports the concept that systemic inflammatory modulation contributes to patients’ overall sense of well-being.

Taken together, these findings indicate that thalassotherapy exerts multidimensional therapeutic effects, simultaneously targeting inflammation, pain, and functional capacity, thereby enhancing the patient’s physical and psychological recovery.

### 3.6. Responder vs. Non-Responder Analysis

To complement the correlation findings, a responder analysis was performed to determine the proportion of patients who achieved clinically meaningful improvements in pain, joint mobility, inflammatory status, and quality of life following the two-week rehabilitation programs (Table 6).

Table 6 shows a higher proportion of responders in the thalassotherapy group than in the standard rehabilitation group, particularly for musculoskeletal outcomes. Clinically meaningful pain reduction was achieved by 68.5% of patients in Group A, compared with 51.7% in Group B (*p* = 0.035). Similarly, mobility improvement ≥ 20% occurred more frequently in the thalassotherapy group (73.3% vs. 57.1%, *p* = 0.028), underscoring its superior functional impact.

The proportion of patients achieving a ≥1 mg/L reduction in CRP was comparable between groups (60.0% vs. 56.7%, *p* = 0.460), suggesting a similar anti-inflammatory response. Quality-of-life improvement ≥10 points was slightly more frequent in Group A (70.0% vs. 63.3%), although this difference did not reach statistical significance (*p* = 0.190).

Overall, these findings demonstrate that thalassotherapy yielded a higher rate of clinically relevant responses in both pain relief and mobility restoration compared with standard rehabilitation, supporting its value as an effective adjunctive strategy in post-traumatic recovery.

## 4. Discussion

The present study demonstrates that a standardized two-week thalassotherapy program produced significantly greater improvements in pain reduction, joint mobility, and quality-of-life outcomes compared with standard rehabilitation in post-traumatic patients. Intragroup analysis (Table 2) confirmed meaningful progress in both groups, but the magnitude of change was markedly higher in patients exposed to marine-based interventions. These findings are consistent with previous balneological and clinical rehabilitation reports from the Black Sea and Mediterranean regions, which have highlighted the therapeutic value of seawater, peloids, and marine aerosols in musculoskeletal recovery [1,2].

Intergroup comparisons (Table 3) revealed that patients receiving thalassotherapy experienced superior musculoskeletal recovery, as evidenced by a greater increase in joint mobility and a more pronounced reduction in pain intensity. The significant decrease in CRP observed in this group suggests that the beneficial effects are not purely mechanical or thermal but also biochemical, reflecting systemic modulation of inflammatory pathways. Similar results have been reported in randomized controlled trials and observational studies, where balneotherapy and mineral-rich mud applications have led to improved functional rehabilitation and decreased inflammatory markers in patients with post-traumatic or degenerative conditions [13,14,15,16].

The reduction of CRP levels observed in the thalassotherapy group provides strong evidence of the anti-inflammatory potential of marine-based therapies. Previous studies have demonstrated that spa and thalassotherapy programs can significantly decrease systemic inflammatory mediators such as CRP and interleukins, while improving endothelial function and vascular elasticity [17,18,19,20]. These data reinforce the concept that thalassotherapy acts on both local and systemic levels—reducing musculoskeletal inflammation, alleviating pain, and contributing to global physiological balance.

The bioactive composition of Techirghiol sapropelic mud may further explain these outcomes. This mud, rich in sulfides, humic substances, and organic compounds, is known to modulate oxidative stress pathways and cytokine profiles, thereby enhancing antioxidant defense and reducing inflammation [1,20,21,22,23]. Such mechanisms likely underlie the consistent decline in CRP observed in our cohort and support the use of mineral-rich peloids as therapeutic agents that extend benefits beyond symptomatic relief.

Recent high-level evidence supports the efficacy of marine-based and balneological therapies, with several meta-analyses demonstrating clinically relevant improvements in pain, mobility, inflammation, and quality of life in musculoskeletal and dermatological conditions. A systematic review and meta-analysis by Shim et al. reported significant reductions in pain and improvements in functional outcomes following marine therapy programs, with effect sizes ranging from moderate to large, depending on the intensity of the intervention [9]. Similarly, De Andrade et al. identified meaningful improvements in pain and physical function following seawater-based aquatic exercise compared with freshwater interventions [21]. More recently, Cegolon et al. synthesized data from controlled trials and concluded that seawater pools provide superior therapeutic effects compared with freshwater pools in rheumatic and inflammatory conditions, particularly through reductions in inflammatory markers and improvements in symptom burden [22,24,25,26]. These meta-analyses provide robust evidence for the therapeutic potential of thalassotherapy under conditions comparable to those evaluated in the present study and underscore the relevance of investigating its effectiveness in post-traumatic rehabilitation.

Improvement in quality of life was another key outcome of the study. Our findings align with those from multicenter investigations showing that balneotherapy significantly enhances psychological well-being, sleep quality, and daily functionality in patients with chronic pain and musculoskeletal disorders [27,28,29]. In our cohort, both rehabilitation modalities improved SF-36 scores; however, thalassotherapy produced a greater magnitude of improvement, emphasizing its holistic effect on physical, emotional, and social dimensions of recovery.

The novelty of this study lies in its comparative approach, directly contrasting thalassotherapy with standard rehabilitation in a homogeneous population of post-traumatic patients. This design allowed the identification of condition-specific advantages attributable to the marine environment, particularly its synergistic biochemical, thermal, and mechanical effects, which cannot be replicated by conventional rehabilitation alone. Although the results consistently favored thalassotherapy, the observational design and short-term assessment limit the strength of causal inferences. Therefore, these findings should be interpreted as early evidence rather than definitive proof of clinical superiority.

Nevertheless, certain limitations should be acknowledged. The relatively short intervention period (two weeks) and the absence of long-term follow-up limit conclusions about the durability of therapeutic gains. Additionally, although the sample size was adequate for the statistical comparisons performed, larger multicenter trials with longer monitoring are warranted to validate and generalize these results.

Overall, the findings of this study suggest that thalassotherapy may offer short-term clinical benefits compared with standard rehabilitation in post-traumatic patients. However, these results should be interpreted cautiously due to the non-randomized design and the limited two-week follow-up. The integration of natural marine resources into rehabilitation programs may support musculoskeletal recovery and symptom relief; however, larger randomized studies with long-term follow-up are needed before confirming the routine clinical integration of thalassotherapy. This study provides the first controlled clinical evidence from Romania comparing standardized thalassotherapy with conventional rehabilitation in post-traumatic patients.

While previous studies have investigated marine-based therapies in mixed or chronic conditions, our work uniquely focuses on acute post-traumatic recovery using a reproducible two-week protocol and integrated outcome measures (clinical, biochemical, and quality-of-life). By integrating seawater baths, sapropelic mud, marine aerosols, and hydrokinetic exercises within a reproducible two-week program, this research provides robust evidence that thalassotherapy not only enhances musculoskeletal recovery, improving joint mobility and pain, but also exerts a systemic anti-inflammatory effect, as reflected by reduced CRP levels. Furthermore, the inclusion of correlation and responder analyses provides novel insights into the physiological mechanisms linking reduced inflammation, functional recovery, and improved quality of life, thereby reinforcing the scientific and clinical relevance of Techirghiol’s marine resources in modern rehabilitation medicine.

Despite its strengths, this study presents several limitations that should be acknowledged. Although the study provides valuable clinical evidence, several limitations should be recognized. The relatively short duration of the intervention and the lack of long-term follow-up restrict the assessment of the sustained effects of thalassotherapy on clinical and inflammatory outcomes. Notably, the absence of medium and long-term follow-up reflects the structural characteristics of the rehabilitation program at the Techirghiol center, where treatment courses are standardized to a two-week schedule and post-intervention evaluations are not routinely implemented within the institutional and regulatory framework; therefore, long-term data could not be collected as part of this study. Moreover, the single-center design may limit the external generalizability of the findings, even though the sample size was adequate for statistical analysis. Additional biochemical or imaging biomarkers, such as cytokine profiles or muscle regeneration indices, were not evaluated; their inclusion could have provided deeper mechanistic insight. Finally, while randomization was not applied due to the observational nature of the study, both groups were carefully matched for age, sex, and baseline functional status to minimize potential selection bias. Although no a priori sample size calculation was performed, as the study included all consecutive eligible patients from a single center, we conducted a post hoc power analysis to verify sample adequacy. With 120 participants, four predictor variables, and α = 0.05, the regression models achieved statistical power above 0.90, indicating sufficient power to detect medium-sized effects. Overall, these limitations do not diminish the clinical relevance of the findings but highlight the need for future multicenter, long-term studies using larger samples and advanced biomarker analyses to confirm and expand the present results.

Future studies should integrate biochemical and molecular biomarkers with AI-based biomechanical monitoring to dynamically evaluate muscle recovery and inflammatory responses during thalassotherapy, paving the way for personalized rehabilitation protocols.

## 5. Conclusions

This study shows that a standardized two-week thalassotherapy program can offer meaningful short-term benefits in post-traumatic rehabilitation, with greater improvements in pain, joint mobility, inflammatory status, and perceived quality of life compared with conventional rehabilitation alone. These effects reflect the combined thermal, mechanical, and biochemical properties of marine factors, which support musculoskeletal recovery and modulate systemic inflammation.

From a practical standpoint, thalassotherapy may be integrated as an adjunctive option in rehabilitation centers, particularly for patients who require enhanced functional recovery or present slow inflammatory resolution. Its good safety profile and high treatment adherence further support its feasibility in clinical practice.

However, the observational design and short follow-up limit the generalizability of the results. Future randomized studies with long-term monitoring are needed to confirm the durability of these improvements and define clear clinical guidelines for incorporating thalassotherapy into routine rehabilitation protocols.

## Figures and Tables

**Figure 1 healthcare-14-00024-f001:**
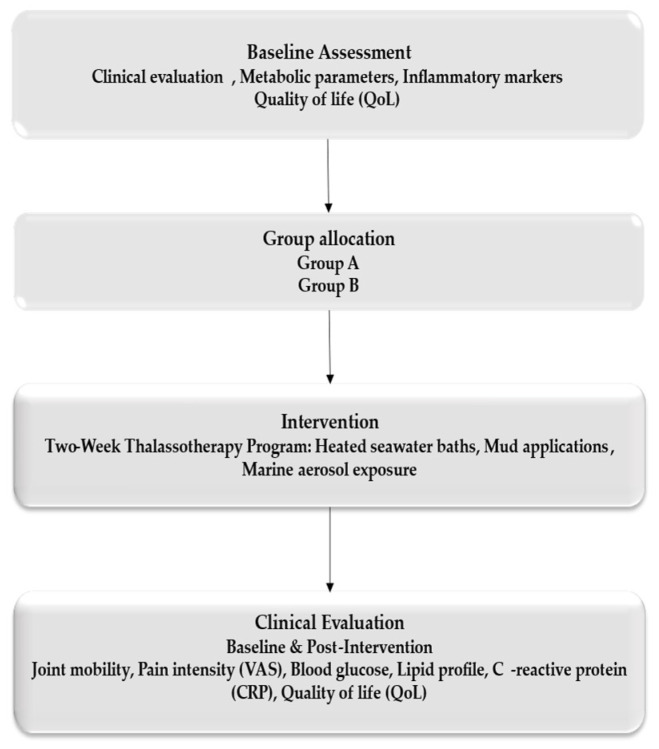
Workflow.

**Table 1 healthcare-14-00024-t001:** Baseline Characteristics.

Characteristic	Group A (n = 60)	Group B (n = 60)	*p*-Value
Age (median, IQR)	46.8 (41–53)	52.4 (48–57)	<0.05
Environment of origin (Urban/Rural)	36 Urban (60.0%)/24 Rural (40.0%)	38 Urban (63.3%)/22 Rural (36.7%)	0.68
Socioeconomic status	Medium: 65%/High: 35%	Medium: 60%/High: 40%	0.52
Educational level	High school: 45%/University: 55%	High school: 48%/University: 52%	0.74

**Table 2 healthcare-14-00024-t002:** Intragroup comparison of clinical and quality of life parameters (baseline vs. post-intervention).

Parameter	Group A—Baseline (Mean ± SD)	Group A—Post (Mean ± SD)	*p*-Value (A)	Group B—Baseline (Mean ± SD)	Group B—Post (Mean ± SD)	*p*-Value (B)
Joint mobility (0–100)	58.2 ± 12.5	75.6 ± 11.4	<0.01	59.0 ± 13.1	66.8 ± 12.2	0.01
Pain (VAS 0–10)	6.8 ± 1.5	3.9 ± 1.4	<0.01	6.5 ± 1.6	5.2 ± 1.5	0.02
CRP (mg/L)	4.6 ± 2.1	3.5 ± 1.8	0.011	4.5 ± 2.0	4.1 ± 1.9	0.14
Quality of life (0–100)	56.0 ± 14.0	67.8 ± 12.5	<0.01	54.5 ± 13.2	61.2 ± 12.8	0.01

Note: Group A = Thalassotherapy (n = 60); Group B = Standard Rehabilitation (n = 60). Statistical significance was set at *p* < 0.05.

**Table 3 healthcare-14-00024-t003:** Intergroup Comparison (Post-Intervention Values).

Parameter	Group A Post (Mean ± SD)	Group B Post (Mean ± SD)	*p*-Value (A vs. B)
Joint mobility (0–100)	72.9 ± 11.8	67.0 ± 11.0	0.01
Pain (VAS 0–10)	4.1 ± 1.6	4.9 ± 1.8	<0.01
Fasting glucose (mg/dL)	101.2 ± 11.5	126.0 ± 20.0	<0.01
Total cholesterol (mg/dL)	192 ± 32	203 ± 38	0.05
CRP (mg/L)	3.8 ± 1.9	4.9 ± 2.3	0.02
Quality of life (0–100)	64.5 ± 13.0	61.0 ± 14.0	0.12

Note: Group A = Thalassotherapy (n = 60); Group B = Standard Rehabilitation (n = 60). Statistical significance was set at *p* < 0.05.

**Table 4 healthcare-14-00024-t004:** Repeated-Measures ANOVA Results.

Parameter	Main Effect: Time (F, *p*)	Main Effect: Group (F, *p*)	Interaction: Group × Time (F, *p*)	Partial η^2^
Pain score (VAS)	F(1, 218) = 51.84, *p* < 0.001	F(1, 218) = 39.17, *p* < 0.001	F(1, 218) = 21.93, *p* < 0.001	0.17
CRP (mg/L)	F(1, 218) = 44.26, *p* < 0.001	F(1, 218) = 30.52, *p* < 0.001	F(1, 218) = 15.66, *p* < 0.001	0.14
Mobility score	F(1, 218) = 48.11, *p* < 0.001	F(1, 218) = 33.48, *p* < 0.001	F(1, 218) = 19.87, *p* < 0.001	0.16
SF-36 total	F(1, 218) = 26.93, *p* < 0.001	F(1, 218) = 20.54, *p* < 0.001	F(1, 218) = 12.88, *p* = 0.001	0.11

**Table 5 healthcare-14-00024-t005:** Correlations Between Clinical Improvement and Inflammatory/Quality-of-Life Parameters (Post-Intervention).

Parameter Correlation	r-Value	*p*-Value
Δ Pain (VAS) − Δ CRP	0.44	0.03
Δ Mobility − Δ Quality of Life	0.56	<0.01
Δ CRP − Δ Quality of Life	0.39	0.04

Note: Δ = change from baseline to post-intervention. Moderate positive correlations indicate that pain reduction and mobility gains were associated with lower CRP levels and improved quality of life.

**Table 6 healthcare-14-00024-t006:** Responder vs. Non-Responder Analysis (Post-Intervention).

Parameter	Group A Responders (%)	Group A Non-Responders (%)	Group B Responders (%)	Group B Non-Responders (%)	*p*-Value (A vs. B)
Pain reduction ≥ 2 VAS points	68.5%	31.5%	51.7%	48.3%	0.035
Mobility improvement ≥ 20%	73.3%	26.7%	57.1%	42.9%	0.028
CRP reduction ≥ 1 mg/L	60.0%	40.0%	56.7%	43.3%	0.460
QoL improvement ≥ 10 points	70.0%	30.0%	63.3%	36.7%	0.190

Note: Responders were defined as patients achieving clinically relevant improvement: ≥20% gain in joint mobility, ≥2-point reduction in VAS pain, ≥1 mg/L reduction in CRP, or ≥10-point increase in quality of life.

## Data Availability

The data supporting the findings of this study are available from the corresponding author upon reasonable request. The dataset contains sensitive medical information protected under GDPR; therefore, it cannot be made publicly available. The data can be provided upon reasonable request to the corresponding author.

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
