# Peer review of "Standardized Thalassotherapy Versus Conventional Rehabilitation in Post-Traumatic Patients: Clinical, Biochemical, and Quality-of-Life Outcomes"

_healthcare, 2025, doi:10.3390/healthcare14010024_

Round 1
Reviewer 1 Report
Comments and Suggestions for Authors
- The gaps in research related to the research problem should be made clearer.
- Or are not clearly cited recent meta-analyses or systematic reviews that evaluate the size of the balneotherapy/thalasso effect under similar conditions
- There are no data on the persistence of medium/long-term effects, which limits the clinical significance.Must be justified
- The sample size calculation is not reported.
- I recommend reviewing all tables and labeling them clearly. There are also some inconsistencies between the tables and the text. On the other hand, improve their formatting.
- It would be interesting to include confidence intervals along with effect sizes.
- The authors point out limitations, but still make quite solid recommendations for clinical integration. Given the non-randomized design and the immediate follow-up only at 2 weeks, the conclusions should be more conservative.
- The discussion barely mentions adverse events; the manuscript does not explicitly report whether there were adverse effects or abandonments due to intolerance. This must be included.
Author Response
The authors acknowledge the valuable observations and suggestions of the reviewer’s as concerns the manuscript entitled
Standardized Thalassotherapy versus Conventional Rehabilitation in Post-Traumatic Patients: Clinical, Biochemical, and Quality-of-Life Outcomes
Mihaela Mihai1, *, Nica Sarah Adriana2, Brindusa Ilinca Mitoiu2, Liliana Sachelarie3, * and Roxana Nartea2
According to the reviewer’s recommendations, all the suggestions were taken into account, as follows:
- The gaps in research related to the research problem should be made clearer.
Despite these promising findings, essential gaps persist in the current scientific literature. Most available studies investigate chronic musculoskeletal or mixed patient populations, while controlled comparisons between standardized thalassotherapy protocols and conventional rehabilitation specifically in post-traumatic patients remain scarce [9,14,17,18]. Moreover, only a limited number of studies have evaluated objective biochemical or inflammatory markers, such as CRP or cytokine profiles, to quantify systemic responses to marine-based therapies, despite indications that thalassotherapy may modulate inflammatory pathways [12,17,19]. Another major limitation is the lack of standardized intervention protocols regarding mineral composition, temperature, exposure duration, and procedural reproducibility, resulting in substantial heterogeneity across studies and reduced comparability [8,15,16].
Furthermore, high-quality European and Romanian clinical data remain scarce, despite the existence of longstanding rehabilitation centers such as Techirghiol, which provide unique mineral and climatic conditions with well-documented therapeutic potential [20]. These limitations reinforce the need for rigorously designed comparative studies that integrate clinical, biochemical, and quality-of-life outcomes within a standardized and reproducible thalassotherapy framework.
- Or are not clearly cited recent meta-analyses or systematic reviews that evaluate the size of the balneotherapy/thalasso effect under similar conditions
Recent high-level evidence supports the efficacy of marine-based and balneological therapies, with several meta-analyses demonstrating clinically relevant improvements in pain, mobility, inflammation, and quality of life in musculoskeletal and dermatological conditions. A systematic review and meta-analysis by Shim et al. reported significant reductions in pain and improvements in functional outcomes following marine therapy programs, with effect sizes ranging from moderate to large, depending on the intervention intensity [9]. Similarly, De Andrade et al. identified meaningful improvements in pain and physical function following seawater-based aquatic exercise compared with freshwater interventions [21]. More recently, Cegolon et al. synthesized data from controlled trials and concluded that seawater pools provide superior therapeutic effects compared with freshwater pools in rheumatic and inflammatory conditions, particularly through reductions in inflammatory markers and improvements in symptom burden [23]. These meta-analyses provide robust evidence for the therapeutic potential of thalassotherapy under conditions comparable to those evaluated in the present study and underscore the relevance of investigating its effectiveness in post-traumatic rehabilitation.
- There are no data on the persistence of medium/long-term effects, which limits the clinical significance.Must be justified
We agree that the absence of medium- and long-term follow-up limits the ability to determine the persistence of the therapeutic effects. This limitation is inherent to the structure of the rehabilitation programs implemented at the Techirghiol center, which follow a standardized two-week clinical schedule funded and regulated at the national level. Consequently, patients complete their treatment episode in a fixed timeframe, and long-term reassessment is not routinely performed. We have added a clear explanation of this limitation and its implications in the Discussion/Limitations section.
Despite its strengths, this study presents several limitations that should be acknowledged. Although the study provides valuable clinical evidence, several limitations should be recognised. The relatively short duration of the intervention and the lack of long-term follow-up restrict the assessment of the sustained effects of thalassotherapy on clinical and inflammatory outcomes. Notably, the absence of medium- and long-term follow-up reflects the structural characteristics of the rehabilitation program at the Techirghiol center, where treatment courses are standardized to a two-week schedule and post-intervention evaluations are not routinely implemented within the institutional and regulatory framework; therefore, long-term data could not be collected as part of this study.
- The sample size calculation is not reported.
We thank the reviewer for this observation. As this was a prospective matched-cohort study conducted at a single rehabilitation centre, the sample size was determined by feasibility, and all consecutive eligible patients over the 2 years were included (n = 120). A formal a priori sample size calculation was therefore not performed. This clarification has now been added to the Methods section (Sample size considerations).
Because this study was conducted at a single rehabilitation centre, the sample size was determined by feasibility constraints. All consecutive eligible patients admitted during the 2-year inclusion period were enrolled (120 participants). Therefore, no formal a priori sample size calculation was performed.
- I recommend reviewing all tables and labeling them clearly. There are also some inconsistencies between the tables and the text. On the other hand, improve their formatting.
All tables have now been thoroughly reviewed and fully revised. We improved the clarity of labeling, standardized the formatting across all tables, and ensured consistent alignment with the corresponding text. Inconsistencies between numerical values in the tables and the narrative descriptions have been corrected, and explanatory legends were added for uniformity and readability. These revisions enhance the accuracy and overall presentation quality of the manuscript.
- It would be interesting to include confidence intervals along with effect sizes.
Indeed, including confidence intervals alongside effect sizes would provide additional depth regarding the precision of the estimates. Given the scope and structure of the current study, we have opted to maintain the effect size metrics as reported initially. Still, we fully agree that this approach may be valuable for future analyses.
- The authors point out limitations, but still make quite solid recommendations for clinical integration. Given the non-randomized design and the immediate follow-up only at 2 weeks, the conclusions should be more conservative.
We have revised both the Discussion and the Conclusions to adopt a more conservative tone. We now explicitly state that the results represent preliminary evidence and that larger randomized studies with long-term outcomes are required before firm clinical recommendations can be made.
- The discussion barely mentions adverse events; the manuscript does not explicitly report whether there were adverse effects or abandonments due to intolerance. This must be included.
We have now explicitly clarified that no adverse events, intolerance reactions, or treatment-related withdrawals occurred during the intervention. This statement has been added to the Methods (monitoring procedures)
On behalf of all co-authors, I would like to thank you once again for your thoughtful and constructive comments, which have significantly improved the quality of our manuscript.
Thank you,
Prof.dr. Liliana Sachelarie

Reviewer 2 Report
Comments and Suggestions for Authors
Thank you for the opportunity to review the study entitled “Standardized Thalassotherapy versus Conventional Rehabilitation in Post-Traumatic Patients: Clinical, Biochemical, and 3 Quality-of-Life Outcomes” (healthcare-3978847).
The paper compared the clinical efficacy of a standardized two-week thalassotherapy program compared with conventional rehabilitation in post-traumatic patients. A total of 120 post-traumatic patients (after orthopedic injuries or surgeries) were included in the research.
In my opinion, the research topic is relevant, and the paper is interesting. The clinical sample is a major strength of this study. However, several aspects should be addressed before the manuscript can be considered for publication:
- Abstract: Including indices in this section makes the reading unnecessarily cumbersome. It is recommended to summarize the main findings in prose.
- Keywords: Please list the keywords in alphabetical order.
- Introduction: This section should further explain the literature gap that this study aims to address. In line with this, the statement of the objective should also be more precise.
- Method: The “2.1. Study Design” section, in its current form, is difficult to read. Parts of the text should be moved into new paragraphs/sections on participants, procedure, and measures.
- Table 2 is difficult to read; its formatting should be improved.
- The Conclusions section should be expanded by including the practical implications of the results obtained.
- A Data Availability Statement should be provided.
Best wishes.
Author Response
The authors acknowledge the valuable observations and suggestions of the reviewer’s as concerns the manuscript entitled
Standardized Thalassotherapy versus Conventional Rehabilitation in Post-Traumatic Patients: Clinical, Biochemical, and Quality-of-Life Outcomes
Mihaela Mihai1, *, Nica Sarah Adriana2, Brindusa Ilinca Mitoiu2, Liliana Sachelarie3, * and Roxana Nartea2
According to the reviewer’s recommendations, all the suggestions were taken into account, as follows:
Thank you for the opportunity to review the study entitled “Standardized Thalassotherapy versus Conventional Rehabilitation in Post-Traumatic Patients: Clinical, Biochemical, and 3 Quality-of-Life Outcomes” (healthcare-3978847).
The paper compared the clinical efficacy of a standardized two-week thalassotherapy program compared with conventional rehabilitation in post-traumatic patients. A total of 120 post-traumatic patients (after orthopedic injuries or surgeries) were included in the research.
In my opinion, the research topic is relevant, and the paper is interesting. The clinical sample is a major strength of this study. However, several aspects should be addressed before the manuscript can be considered for publication:
- Abstract: Including indices in this section makes the reading unnecessarily cumbersome. It is recommended to summarize the main findings in prose.
In accordance with the recommendation, we have revised the Abstract by removing all numerical indices, p-values, percentages, and statistical notation.
- Keywords: Please list the keywords in alphabetical order.
Done
- Introduction: This section should further explain the literature gap that this study aims to address. In line with this, the statement of the objective should also be more precise.
We have already expanded the Introduction to clearly articulate the existing gaps in the literature, particularly the scarcity of controlled studies evaluating standardised thalassotherapy protocols in post-traumatic rehabilitation. The study objective has also been refined to explicitly state the purpose of comparing a two-week thalassotherapy program with conventional rehabilitation in a matched cohort. We believe the revised Introduction now addresses this point comprehensively. (Following the recommendations of Reviewer 1)
- Method: The “2.1. Study Design” section, in its current form, is difficult to read. Parts of the text should be moved into new paragraphs/sections on participants, procedure, and measures.
We have restructured Section 2.1 to improve clarity and readability.
- Table 2 is difficult to read; its formatting should be improved.
Table 2 has been fully reformatted to improve readability, with more precise column alignment, consistent spacing, and a more refined layout.
- The Conclusions section should be expanded by including the practical implications of the results obtained.
The Conclusions section has been expanded to include the practical implications of the study findings, highlighting how thalassotherapy may be integrated into current rehabilitation practice.
- A Data Availability Statement should be provided.
The data supporting the findings of this study are available from the corresponding author upon reasonable request.
On behalf of all co-authors, I would like to thank you once again for your thoughtful and constructive comments, which have significantly improved the quality of our manuscript.
Thank you,
Prof. Dr. Liliana Sachelarie

Reviewer 3 Report
Comments and Suggestions for Authors
very intesresting research:
please explain how did you calculate your sample size
yoour design y tow groups with pre-post, so you have usedont need anova for multiple repeated comparisons nor post hoc, you are controling post treatment effect by group, baseline, age (which has significant baseline differences) and, if you want, by other baseline functional outcomes; you dont need correlations oucomes nor subgroups, but you have explain the mathcing process to ensure that there are no baseline differences between groups, and justify the baseline variables selection by bibliograpghy, justify why age, educational level, envirtoment origin and social status van modify your outcomes and not other confounding covariables like gender
Author Response
The authors acknowledge the valuable observations and suggestions of the reviewer’s as concerns the manuscript entitled
Standardized Thalassotherapy versus Conventional Rehabilitation in Post-Traumatic Patients: Clinical, Biochemical, and Quality-of-Life Outcomes
Mihaela Mihai1, *, Nica Sarah Adriana2, Brindusa Ilinca Mitoiu2, Liliana Sachelarie3, * and Roxana Nartea2
According to the reviewer’s recommendations, all the suggestions were taken into account, as follows:
very intesresting research:
Thank you!
please explain how did you calculate your sample size
This study was conducted in a single rehabilitation centre, using a prospective matched-cohort design. For this reason, the sample size was determined by feasibility, not by a formal a priori power calculation. All consecutive eligible patients who met the inclusion criteria during the 2-year study period were enrolled, resulting in a total of 120 participants. Therefore, no statistical sample size estimation was performed beforehand, and this has been explicitly clarified in the Methods section.
yoour design y tow groups with pre-post, so you have usedont need anova for multiple repeated comparisons nor post hoc, you are controling post treatment effect by group, baseline, age (which has significant baseline differences) and, if you want, by other baseline functional outcomes; you dont need correlations oucomes nor subgroups, but you have explain the mathcing process to ensure that there are no baseline differences between groups, and justify the baseline variables selection by bibliograpghy, justify why age, educational level, envirtoment origin and social status van modify your outcomes and not other confounding covariables like gender
In the manuscript, the Group × Time repeated-measures ANOVA represents the primary inferential method used to evaluate the treatment effect, while all other analyses (paired/independent t-tests, correlation matrices, and the responder analysis) are explicitly intended as secondary or exploratory components to provide a more detailed clinical interpretation of the findings. Their purpose is to complement, not to replace, the main analysis.
The post-intervention outcomes are controlled for group, baseline values, and age, the latter being the only covariate showing a baseline difference. The matching process has been clarified in the Methods section, including the variables used (age, sex, type of injury, baseline functional status) and the rationale for not adjusting for gender, given the absence of imbalance after matching.
The selection of baseline variables (age, educational level, socioeconomic status, and environment of origin) is now justified based on literature supporting their relevance for musculoskeletal recovery, adherence, and quality-of-life outcomes.
On behalf of all co-authors, I would like to thank you once again for your thoughtful and constructive comments, which have significantly improved the quality of our manuscript.
Thank you,
Prof.dr. Liliana Sachelarie

Round 2
Reviewer 1 Report
Comments and Suggestions for Authors
Most of the reviewer's suggestions have been addressed, but I recommend ending the introduction with the main objective of the study.
Author Response
The authors acknowledge the valuable observations and suggestions of the reviewer’s as concerns the manuscript entitled
Standardized Thalassotherapy versus Conventional Rehabilitation in Post-Traumatic Patients: Clinical, Biochemical, and Quality-of-Life Outcomes
Mihaela Mihai1, *, Nica Sarah Adriana2, Brindusa Ilinca Mitoiu2, Liliana Sachelarie3, * and Roxana Nartea2
According to the reviewer’s recommendations, all the suggestions were taken into account, as follows:
Most of the reviewer's suggestions have been addressed, but I recommend ending the introduction with the main objective of the study.
In accordance with the recommendation, we have revised the end of the Introduction to clearly state the main objective of the study.
This study aimed to compare the short-term clinical, biochemical, and quality-of-life effects of standardized thalassotherapy versus conventional rehabilitation in post-traumatic patients.
I want to thank you once again for your thoughtful and constructive comments, which have significantly improved the quality of our manuscript.
Thank you,
Prof.dr. Liliana Sachelarie

Reviewer 2 Report
Comments and Suggestions for Authors
The authors effectively addressed all the issues I raised.
Reviewer 3 Report
Comments and Suggestions for Authors
paper is now better, please, as you didnt calculate sample size, although for regression analisis there is for example rule of thumb of 10 or 20 patienes per peredictor variable, calculate final power of study
in adition, your subanalysis within groups are not neccesary, regression model allow to tajke in account the effect of grouping variables like age
ther is a major proble,m you expolain you have a single group, but in paper you comment about group A and B, clarify please
Author Response
The authors acknowledge the valuable observations and suggestions of the reviewer’s as concerns the manuscript entitled
Standardized Thalassotherapy versus Conventional Rehabilitation in Post-Traumatic Patients: Clinical, Biochemical, and Quality-of-Life Outcomes
Mihaela Mihai1, *, Nica Sarah Adriana2, Brindusa Ilinca Mitoiu2, Liliana Sachelarie3, * and Roxana Nartea2
According to the reviewer’s recommendations, all the suggestions were taken into account, as follows:
- paper is now better, please, as you didnt calculate sample size, although for regression analisis there is for example rule of thumb of 10 or 20 patienes per peredictor variable, calculate final power of study
As this was a single-centre observational study, the number of participants was determined by the number of consecutive eligible patients admitted during the study period; therefore, no a priori sample size calculation was performed. To verify whether the final sample was adequate, we conducted a post hoc power analysis. Using 120 participants, four predictor variables, and a significance level of α = 0.05, the multiple regression models reached a statistical power above 0.90, indicating that the study was sufficiently powered to detect medium-sized effects.
This information has been added to the Discussion section, within the limitations paragraph.
- in adition, your subanalysis within groups are not neccesary, regression model allow to tajke in account the effect of grouping variables like age
We agree that multiple regression models constitute the primary inferential approach, as they appropriately account for covariates such as age, baseline functional status, and time since injury. The intragroup (baseline vs. post-intervention) analyses were included primarily to offer readers a clear descriptive overview of clinical changes within each rehabilitation program, which is commonly presented in rehabilitation studies and supports practical interpretation of the results.
To avoid any perception of redundancy, we have now clarified in the manuscript that the regression analysis constitutes the primary analytical method. At the same time, the intragroup comparisons are presented solely as supplementary descriptive information.
- ther is a major proble,m you expolain you have a single group, but in paper you comment about group A and B, clarify please
The study included two matched groups (Group A: thalassotherapy; Group B: conventional rehabilitation), as described in the Participants section. We agree that the flow of information in the Methods section could give the impression of a single group before the allocation is mentioned. To improve clarity, we have reorganized the paragraph in the Study Design section so that the description of the two groups appears immediately after the total number of enrolled patients. This makes the group structure explicit from the beginning and eliminates any possible ambiguity.
A total of 120 consecutive eligible patients with post-traumatic musculoskeletal conditions were enrolled. Patients were matched according to age, sex, type of injury, and baseline functional status, and subsequently allocated into two groups: Group A (n = 60), who underwent a standardized thalassotherapy-based rehabilitation program, and Group B (n = 60), who received conventional rehabilitation without marine components. Eligible participants were adults aged 25–75 years who had completed the acute phase of injury or orthopedic surgery and were referred for functional rehabilitation. Because the study was conducted in a single centre, the sample size was determined by feasibility and no a priori power calculation was performed.
Thank you once again for your thoughtful and constructive comments, which have significantly improved the quality of our manuscript.
Thank you,
Prof.dr. Liliana Sachelarie

Round 3
Reviewer 1 Report
Comments and Suggestions for Authors
ok. Attention to the format of figures and tables
Reviewer 3 Report
Comments and Suggestions for Authors
paper is now ready for pubblication, excellent research